# The Hydrolysis of Phosphinates and Phosphonates: A Review

**DOI:** 10.3390/molecules26102840

**Published:** 2021-05-11

**Authors:** Nikoletta Harsági, György Keglevich

**Affiliations:** Department of Organic Chemistry and Technology, Budapest University of Technology and Economics, 1521 Budapest, Hungary; harsagi.nikoletta@vbk.bme.hu

**Keywords:** hydrolysis, dealkylation, phosphinates, phosphonates, *P*-acids

## Abstract

Phosphinic and phosphonic acids are useful intermediates and biologically active compounds which may be prepared from their esters, phosphinates and phosphonates, respectively, by hydrolysis or dealkylation. The hydrolysis may take place both under acidic and basic conditions, but the C-O bond may also be cleaved by trimethylsilyl halides. The hydrolysis of *P*-esters is a challenging task because, in most cases, the optimized reaction conditions have not yet been explored. Despite the importance of the hydrolysis of *P*-esters, this field has not yet been fully surveyed. In order to fill this gap, examples of acidic and alkaline hydrolysis, as well as the dealkylation of phosphinates and phosphonates, are summarized in this review.

## 1. Introduction

Phosphinic and phosphonic acids are of great importance due to their biological activity (Figure 1) [1]. Most of them are known as antibacterial agents [2,3]. Multidrug-resistant (MDR) and extensively drug-resistant (XDR) pathogens may cause major problems in the treatment of bacterial infections. However, Fosfomycin has remained active against both Gram-positive and Gram-negative MDR and XDR bacteria [2]. Acyclic nucleoside phosphonic derivatives like Cidofovir, Adefovir and Tenofovir play an important role in the treatment of DNA virus and retrovirus infections [4]. Some *P*-esters have also been shown to be effective against Hepatitis C and Influenza A virus [5], and some are known as glutamate and GABA-based CNS therapeutics [5,6,7]. Glutamate is a main excitatory neurotransmitter, so agonists of the metabotropic glutamate receptor can be new therapeutic targets for brain disorders (schizophrenia, Parkinson’s disease, pain). GABA is a main inhibitory neurotransmitter which is responsible for neurological disorders (epilepsy, anxiety disorders). Dronates are known to increase the mineral density in bones [8,9]. Moreover, *P*-esters include antimalarial agents [5,10,11], anticancer agents [5,12,13,14] and angiotensin-converting enzyme (ACE) inhibitors [15]. In addition, the use of *P*-acids as herbicides (glyphosate, glyfosinate) [5,16] is not negligible either. Phosphinic acids are of interest due to their ability to inhibit metalloproteases [17]. Methylphosphonic acid is known as a flame retardant [18]. During the preparation of these compounds, an ester-protecting group is introduced into the molecule, and the hydrolysis of the ester group is necessary in the final steps.

For a long time, water was used as the solvent only in hydrolyses. Despite its favorable properties (cheap, available, safe and “green”), water could not spread as a general solvent. This is due to the low solubility of organic substrates. The application of co-solvents, such as alcohols, DMF, acetone and acetonitrile is a good possibility. However, the regeneration of water or water–solvent mixtures is not easy.

Despite their great importance, the hydrolysis of *P*-esters has not been adequately studied. Often, unoptimized routine hydrolyses were described or the kinetics of these processes were studied. In most cases, the esters were reacted under harsh conditions with a large excess of concentrated acid, and often the applied reaction time was longer than necessary. Hydrolyses can be catalyzed by acids and bases as well (Scheme 1) [19]. Acidic hydrolyses can be catalyzed both by mineral and Lewis acids [20]. Mineral acids are mostly hydrogen halides [21], e.g., hydrochloric acid [22,23,24,25,26,27], but hydrobromic acid [28,29,30,31,32] proved to be more efficient. Despite this, the application of hydrochloric acid was widespread. The hydrolyses were generally performed at around 100 °C with longer reaction times [20]. There are also examples when trifluoroacetic acid (TFA) [17,33,34] or HClO_4_ [35,36] was used to catalyze the hydrolyses. Sodium hydroxide is the most commonly used reagent in alkaline hydrolysis [37,38,39,40,41], but there are also examples of the application of KOH [42], LiOH [40,43] and NaHCO_3_ [44]. The alkaline hydrolysis is irreversible and less corrosive, but alkali-sensitive molecules can be damaged. A further disadvantage is that the base-catalyzed hydrolyses take place in two steps: first, the sodium-salt of the acid is formed, then the corresponding acid is liberated. In the case of acid catalysis, the *P*-acids are obtained directly.

During hydrolysis, a nucleophilic attack occurs on the phosphorus atom of the P=O unit [45]. In most of the cases, the P-O bond is cleaved during the acid- and base-catalyzed hydrolyses. The rate of hydrolysis may be influenced by the pH [35,36] and by the ionic strength of the medium, but the type of ion added to the system is also decisive.

There are a few cases when the desired *P*-acid is not prepared by hydrolysis, but by the cleavage of the C-O bond, which is possible by pyrolysis [46] or in reaction with trimethylsilyl halides (Scheme 1) [47,48,49,50,51,52], boron tribromide [53], or various amines [54,55,56]. Dealkylations with trimethylsilyl halides takes place under mild conditions, such that they can also be used for the hydrolysis of esters in which instances strong acidic or alkaline treatments cannot be applied, such as in the cases of nitriles, vinyl ethers and acetals [57,58,59]. In addition, various enzyme-catalyzed [60,61,62,63,64,65,66,67,68,69,70,71,72] hydrolyses have also been elaborated, and there are examples for the application of special catalysts and metal ions as well [73,74,75,76,77].

It is important to mention that *P*-acids can also be prepared indirectly (Scheme 2). In this case, the ester (**1**) is converted to the corresponding acid chloride (**2**), which is a more reactive derivative, and can react with water at room temperature [78,79]. This method cannot be considered as a good solution from the point of view of its number of steps and atomic efficiency.

In this survey, we discuss the acidic hydrolysis of phosphinates and phosphonates. This is followed by the presentation of the alkaline and basic hydrolysis of phosphinates and that of phosphonates. The reactivity of the different substrates, the effect of the substituents, and their green chemical aspects are the focus. Last but not least, the conversion of *P*-esters to acids by dealkylation is summarized.

## 2. Acidic Hydrolysis of Phosphinates and Phosphonates

### 2.1. Acidic Hydrolysis of Phosphinates

In a paper published in 1973, Cook et al. compared the acid-catalyzed hydrolysis of methyl dialkylphosphinates (**4**) with base-catalyzed examples (Scheme 3) [80]. It was concluded that polar and steric effects hardly influence the acid-catalyzed hydrolysis compared to the base-catalyzed version. In a subsequent publication, they demonstrated that the hydrolysis of the methyl esters proceeds by the rarely occurring A_Al_2 mechanism (water is involved and C-O bond cleavage occurs) [81]. The major routes involve the A_Ac_2 mechanism (water is involved and P-O bond cleavage occurs) and the A_Al_1 mechanism (water is not involved in the rate-determining step, and a C-O bond cleavage occurs) [82]. The mechanistic study was extended to the hydrolysis of additional esters [82,83].

Bunnett et al. studied the hydrolysis of different methyl methyl-arylphosphinates (**6**) at various HClO_4_ concentrations (1–9 M) and temperatures (67.2, 95.1, 107.6 °C) (Scheme 4) [35]. The hydrolysis was found to be optimal at a 6–7 M acid concentration, above which the reaction became slightly slower.

The hydrolysis of *p*-nitrophenyl diphenylphosphinate (**8**) under acid catalysis was also studied (Scheme 5) [36]. The rate constant was determined at different acid concentrations in a dioxane–water mixture to ensure homogeneity. There is a maximum rate at 1.5 M HClO_4_. In more concentrated solutions, the acidic inhibition of the hydrolysis was observed.

There were cases when alkaline hydrolysis proved to be slow at room temperature, but at higher temperatures it was too harsh for sensitive substrates. In these cases, acidic hydrolysis is more favorable. A good example is the acidic hydrolysis of a β-carboxamido-substituted phosphinic acid ester (**10**), as this is a rapid and gentle way to provide the corresponding phosphinic acid (**11**) quantitatively (Scheme 6) [17]. In this particular case, trifluoroacetic acid was the catalyst in an aqueous medium.

In the following example, the preparation of bis(3-aminophenyl)phosphinic acid (**13**) using hydrochloric acid as the catalyst in ethanol is demonstrated (Scheme 7) [84]. The starting bis(aniline) derivative was obtained from the bis-nitro compound by reduction. The exact conditions were not reported.

α-Aminophosphinic acids and their derivatives form an important group due to their synthetic and medicinal interest. The hydrolysis of phosphinates **14** using *cc*. HCl (Scheme 8) [85] or HBr/AcOH [86] provided optically active α-aminophosphinic acids (**15**).

The hydrolysis of β-aminophosphinates (**16**) was performed using hydrochloric acid at the boiling point (Scheme 9) [87]. To 3 g substrate, 20 mL cc. HCl was added, and the mixture was stirred at reflux for 1.5–4 h.

The hydrolysis of a GABA_B_ antagonist ethyl phosphinate (**18**) was performed by applying *cc*. HCl at 100 °C for 24 h (Scheme 10) [6].

Natchev and co-workers investigated the acidic and enzymatic hydrolysis of phosphoryl analogues of glycine [61]. While the acidic hydrolysis was performed using 15–20% acid at reflux for 6–7 h to afford the corresponding phosphinic acid (**21**) at a yield of 94%, the enzymatic variation carried out using α-chymotrypsin under milder conditions (37 °C for 6 h) gave the acid quantitatively (Scheme 11).

During the preparation of β-functionalized hydroxymethylphosphinic acid derivatives (**22**), double hydrolysis took place at a temperature of 80 °C in 3 h. In this case, 15 equivalents of 35% hydrochloric acid were used, which may be regarded as a large excess (Scheme 12) [88].

Dennis and co-workers investigated the hydrolyses of different saturated and unsaturated cyclic phosphinates, along with their open chain analogues under acidic conditions (Figure 2) [89]. They compared the rate constants of similar derivatives, and found the following ratios: k_A_/k_D_ 1; k_B_/k_E_ 1; k_C_/k_F_ 3. These results are surprising, as the hydrolysis of cyclic phosphonates and phosphates is generally much faster than that of their open-chain analogues.

The most general procedure to prepare phosphinic acids from their esters involves the use of a concentrated HCl solution at reflux. Different GABA_Cρ_ antagonists, such as 1-hydroxyphospholane oxide **25**, were prepared by hydrolysis with hydrochloric acid [7]. To 0.50 mmol cyclic ester **24**, 2 mL HCl was added, and the mixture was refluxed for 5 h (Scheme 13).

In our research group, the hydrolysis of a series of cyclic phosphinates (**26**) was investigated (Scheme 14) [90]. First, we wished to explore the optimal reaction conditions, including the reaction time, acid concentration, and the necessary amount of hydrochloric acid. It was found that for the hydrolysis of 1.9 mmol phosphinate (**26**), the use of 0.5 mL *cc*. HCl and 1 mL water was optimal, along with a reaction time of 6 h. Interestingly, in the case of the hydrolysis of unsaturated cyclic phosphinates, these compounds underwent isomerization as well. The results are summarized in Table 1.

After exploring the optimum conditions, the method was extended to the hydrolysis of other esters, and the kinetics were also investigated. Figure 3 demonstrates the order of reactivity observed under the above-mentioned conditions [90].

The hydrolyses were also carried out under microwave (MW) conditions. In this case, *p*-toluenesulfonic acid (PTSA) was used as the catalyst in order to avoid the corrosion of the reactor (Scheme 15) [90]. It is important to note that due to the beneficial effect of MW irradiation, the reaction times were shorter than they were in the case of conventional heating.

The acidic hydrolysis of acyclic esters, such as diphenylphosphinates (**30**), was also studied under conventional heating and microwave irradiation (Scheme 16) [91]. The traditional hydrolysis was performed using three equivalents of diluted hydrochloric acid for 3–6.5 h. In the other series of experiments comprising MW-assisted hydrolyses, *p*-toluenesulfonic acid was used as the catalyst. The amount of the catalyst was decreased to 0.1 equivalents. At 160 °C, complete hydrolysis occurred in 2–6.5 h, and at 180 °C in 0.5–2 h. The pseudo–first-order rate constants obtained are listed in Table 2.

### 2.2. Acidic Hydrolysis of Phosphonates

The hydrolysis of phosphonates is a widely applied method. Due to the two ester groups, these hydrolyses take place in two steps in a consecutive manner. Most often, the aqueous solution of hydrochloric acid was applied as the medium, and after the hydrolysis, the water was removed by distillation. Occasionally, HBr was also used in the hydrolysis of phosphonates [21].

Methylphosphonic acid (**32**), which is known as a flame retardant, may be prepared by the acidic hydrolysis of dimethyl methylphosphonate (**31**) (Scheme 17) [18].

The effect of the alkyl group of dialkyl phosphonates was investigated in acid- and base-catalyzed hydrolyses. It was found that during acid catalysis, the isopropyl derivative was hydrolyzed faster than the methyl ester, but under basic conditions, the reaction of the methyl ester was 1000-fold faster than that of the isopropyl derivative [92].

Our research group investigated the preparation of arylphosphonic acids (**34**) by refluxing the corresponding phosphonates (**33**) with an excess (six equivalents) of hydrochloric acid for 12 h (Scheme 18) [93].

Depending on the substituents, the phosphonic acids were obtained in yields of 71–93% [93]. However, a reflux with concentrated hydrochloric acid for 12 h cannot be considered to be a “gentle” method.

We also studied the hydrolysis of various arylphosphonates (**35**): phenylphosphonates and their derivatives containing a 4-methyl or a 4-acetyl group in the phenyl ring. The reactions were carried out at the optimum conditions found for the hydrolysis of cyclic phosphinates (Scheme 19) [94]. In most of the cases, the reaction proceeded according to the A_Ac_2 mechanism, but in the case of the benzyl and isopropyl ester, the A_Al_1 mechanism was substantiated.

The consecutive reaction steps were characterized by pseudo–first-order rate constants. One can see that the cleavage of the second P-OC bond was the slower process in each case, suggesting that the latter is the rate-determining step (Table 3).

Based on the experimental data, the overall reactivity order of the different derivatives (R^2^/R^1^) is the following:Bn/H >> ^i^Pr/H ~Me/H > Et/C(O)Me > Et/H > Et/Me.

Biologically important α-hydroxyphosphonic acids (**38**, **40**) were prepared by the prolonged (1–2 days) heating of various hydroxyphosphonates with a large excess of hydrochloric acid (Scheme 20) [95].

α-Hydroxyphosphonic acids (**42**) were prepared by the hydrochloric acid-promoted hydrolysis of hydroxyphosphonates (**41**) (Scheme 21) [96]. The hydrolysis was performed using 6 N HCl in dioxane-water at 80 °C for 3 days.

We investigated the acid-catalyzed reactions of a series of α-hydroxy-benzylphosphonates (**43**) in order to evaluate the effect of the ester function, and the substituents in the phenyl ring on the rate (Scheme 22) [97]. The reactions were performed in water with 3 equivalents of hydrochloric acid, and depending on the substituents, the completion took 2.5–9.5 h. Electron-withdrawing substituents increased the reaction rate, while electron-releasing substituents slowed down the hydrolysis. The experimental and kinetic data are summarized in Table 4, while representative concentration profiles of the hydrolyses are shown in Figure 4.

Aminophosphonic acids form another important family of bioactive compounds. These species are the analogues of amino acids. The preparation of aminomethylphosphonic acid (**47**) by hydrolysis involved the removal of the protecting group (Scheme 23) [3]. 

There is also an example for the HBr/acetic acid-catalyzed hydrolysis of aminophosphonates [98]. In the synthesis of a biologically relevant aminomethylene-bisphosphonic acid (**49**), the last step involved a HCl-catalyzed hydrolysis (Scheme 24) [99].

Phosphonic acid analogues of certain amino acids may have significant biological effects, e.g., arginine mimetics inhibit the activity of the enzymes responsible for the survival of parasites; hence, they can be used as anti-malarial agents [100]. The corresponding compounds were obtained by hydrolysis with hydrochloric acid [100], or acetic acid combined with hydrogen bromide [31]. This change in the functionality was performed in the last step of the synthesis. Similar compounds were prepared from thioureidoalkane phosphonates by treatment with acetic acid and hydrochloric acid at reflux for 7 h [101].

The hydrolysis of a benzimidazole phosphonate (**50**) was carried out using a 40% HBr solution at reflux for 10 h (Scheme 25) [28].

In the following example, the hydrolysis of the succinic acid derivative (**52**) took place in an autocatalytic manner. The two succinate functions were also hydrolyzed under the conditions applied. The intermediates may catalyze further hydrolysis due to their acidic nature (Scheme 26) [79]. The ethanol released was removed by azeotropic distillation.

Following the spread of the MW technique, the effect of irradiation on hydrolysis was also studied [20,102]. An example from the pharmaceutical field is the synthesis of Adefovir, during which the diisopropyl ester moiety of phosphonate **54** was hydrolyzed in an acid-catalyzed manner under MW conditions (Scheme 27) [20].

The acidic hydrolysis of a series of alkyl α-hydroxyimino-α-(*p*-nitrophenyl) alkylphosphonates (**56**) revealed that the reaction rate decreases with increasing steric hindrance (Scheme 28) [103]. In addition to the ester function, the neighbouring groups also had a significant effect on the hydrolysis, e.g., when a *tert*-butyl group (R^2^) was replaced by a methyl substituent, a 100-fold reaction rate was observed [103].

## 3. Alkaline and Basic Hydrolysis

### 3.1. Alkaline and Basic Hydrolysis of Phosphinates

The effect of various factors on alkaline hydrolysis was investigated in a few publications [104,105,106]. The influencing factors include the leaving ability of the departing group [104,105,107,108], the stability of the resulting intermediate [107], the nature of the heteroatom connected to the phosphorus atom [106], the solvent [106], the pH [109] and the temperature applied, all of which may have a significant effect on the course of the hydrolysis.

Two equivalents of NaOH in water were used in the hydrolysis of a series of ethyl phosphinates (**58**) carried out with stirring at 80 °C for 6–12 h. The sodium salt formed during the alkaline hydrolysis was converted to free acid (**5**) by treatment with hydrochloric acid in the second step (Scheme 29) [110].

The steric effects play a significant role [111]. The alkaline hydrolysis of the ethyl diethyl, diisopropyl and di-*tert*-butyl phosphinates (**58**) was studied (Scheme 30). It was found that the increase in the steric hindrance decreased the reaction rate significantly [111]. Relative rate constants of 260, 41 and 0.08 were reported for the alkaline hydrolysis of the diethyl ester (at 70 °C), the diisopropyl ester (at 120 °C) and the di-*tert*-butyl ester (at 120 °C). 

In the alkaline hydrolysis of sterically hindered phosphinates (**59**), ethyl di-*tert*-butylphosphinate hydrolyzed 500 times slower than ethyl diisopropylphosphinate (Scheme 31) [112]. The major factors influencing the alkaline hydrolysis of the *P*-esters are the steric hindrance within the phosphinate and the strength of the acid resulting from the hydrolysis.

During the hydrolysis of 1-alkoxyphospholene oxides, it was found that, under alkaline conditions, the 1-alkoxy-3-phospholene oxide hydrolyzed 40 times faster than the 1-alkoxy-2-phospholene oxide [104]. In another paper, a series of methyl esters (**4**) (Scheme 32) and other cyclic phosphinates were investigated in order to clarify the effect of alkyl groups and rings [113]. In this case, the reaction conditions and the yields were not provided.

The order of reactivity for the open chain derivatives was the following [113]:Me > Ph > Bn > Et > ^n^Bu

In the case of cyclic derivatives, the following order of reactivity was found [113]:



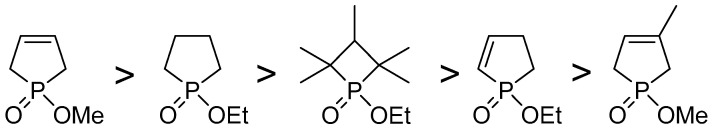



The alkaline hydrolysis of other cyclic and open-chain phosphinates was also studied [114] and the rate constants were determined. In the case of less-soluble phosphinates, alcohol–water mixtures were used. The hydrolysis of five-membered cyclic phosphinates was faster than that of the open-chain and six-membered cyclic phosphinates (Table 5).

The effect of various heteroaromatic substituents was also studied in the hydrolyses carried out in 50% aqueous dioxane: the hydrolysis of di(2-furyl)-, di(2-thienyl)- and diphenylphosphinic acid ethyl esters performed using one equivalent of sodium hydroxide for 72–120 h was compared (Figure 5) [115]. It was found that the hydrolysis of the furyl derivative was the fastest, while that of the phenyl species was the slowest.

Bergesen investigated the difference between the alkaline hydrolysis of the *cis* and *trans* isomers of a four-membered cyclic phosphinate (**60**) in 50% ethanol–water (Scheme 33) [39]. The results showed that the *cis* isomer hydrolyzed ca. seven times faster. This may be explained by the fact that, in the case of the *trans* isomer, the three methyl groups block the access of the hydroxy ion, as compared to the case when only two methyl groups are in the neighborhood, as in the *cis* isomer.

The alkaline hydrolysis of different alkyl diethylphosphinates (**59**) was also investigated (Scheme 34) [38]. The reactivity of the alkoxy groups was influenced by the steric bulk of the alkyl group. The value of the rate constants decreased with the increase of the steric hindrance. However, the exact conditions and outcomes were not reported.

It was confirmed by another group that the base-catalyzed hydrolysis became less efficient with the increasing steric requirement of the alkyl group of the alkoxy moiety [104].

Furthermore, 1-Hydroxy-3,4-diphenylphosphole-1-oxide (**63**) was prepared from the corresponding phenoxyphosphole oxide (**62**) by alkaline hydrolysis (Scheme 35) [105]. Then, the corresponding phosphinic acid (**61**) was liberated with HCl.

Studying the hydrolysis of different esters, Clarke and co-workers concluded that the smaller the electron-releasing effect of the substituents was, the greater the rate constant became (Figure 6) [105].

The NaOH-catalyzed hydrolysis of substituted aryl diphenylphosphinates (**64**) was also investigated (Scheme 36) [106]. It was found that the value of the rate constant decreased with the decrease of the electron-withdrawing ability of the substituent Y in the departing aryl ring. All of the reactions were carried out under pseudo–first-order kinetic conditions, but the exact circumstances were not reported. 

Hydrolyses of various aryl diphenylphosphinates (**64**) carried out using OH^–^ and imidazole catalysis were compared (Scheme 37) [116]. In addition to the substituent dependence, it was found that the imidazole-promoted hydrolyses were significantly faster than the OH^–^-catalyzed examples. This paper was merely a kinetic study, and the exact conditions were not reported.

The effect of alkyl groups was also studied in the case of diphenylphosphinothioates. It was found that the electron-withdrawing effect accelerates the process, whilst the electron-releasing effect greatly slows it down [117]. 

It was also found that thioesters (>P(O)SR) are much more reactive than the oxo analogues. The reason is that the RS substituent is a better leaving group. In addition, the R group has a greater influence on the hydrolyzing ability of OR than it does on that of SR [106,118]. Comparing the reactivity of the P=O and P=S derivatives, it can be said that in the case of alkaline hydrolysis, the oxo derivatives are more reactive. [106,109].

The effect of solvents and solvent mixtures was also studied. It was found that the hydrolysis was slightly faster in solvent mixtures [106]. Possible solvent mixtures may be 60% dimethoxyethane in water [104,114], 60% dimethyl ether in water [106], 20% acetonitrile in water [119], and 60% acetone in water [117], but there are other options as well, e.g., hydrolysis in dioxane–water [117] or in methanol–water [117].

### 3.2. Alkaline and Basic Hydrolysis of Phosphonates

The alkaline hydrolysis of a series of diethyl alkylphosphonates (**65**) was investigated in DMSO/H_2_O. Based on the results, an order of reactivity was established on the basis of the nature of the various alkyl chains (Scheme 38) [38]. Higher reactivity was observed for the esters with an *n*-alkyl substituent, while the rate of the hydrolysis decreased with increasing steric hindrance. 

The steric effects had a greater influence on the hydrolysis of phosphonates compared to that of carboxylic esters. In addition, it was found that the hydrolysis of six- and seven-membered cyclic phosphonates is faster than that of the open-chain analogues [38].

It was observed that the rate of the hydrolysis was greatly influenced by the nature of the leaving group and the substituents on the phosphorus atom. Aksnes et al. studied the alkaline hydrolysis of various diethyl alkyl-, chloromethyl- and dichloromethylphosphonates (**65**) in an acetone–water solvent-mixture (Scheme 39) [120]. The presence of the chloromethyl or dichloromethyl substituents increased the reaction rate. Compared to the hydrolysis of carboxylic esters, the hydrolysis of phosphonates is less sensitive to electronic effects.

As an interesting example, a diphenyl adenosilvinylphosphonate (**67**) was hydrolyzed in the presence of ammonium fluoride (Scheme 40) [121].

Other vinylphosphonic esters (**69**) were hydrolyzed under similar conditions (Scheme 41) [121]. 

It was noted that, in the case of benzyl esters, the corresponding acids may also be obtained by catalytic hydrogenation.

The enzyme-catalyzed hydrolysis of diphenyl alkylphosphonates (**71**) was also reported [121,122]. As a matter of fact, the hydrolysis of the first ester function was performed by applying base catalysis, while a phosphodiesterase enzyme was used in the second step (Scheme 42).

The above phenomenon was investigated by several groups. Hudson et al. also studied the effect of the *P*-substituents on the reactivity [92]. 

In the acidic hydrolysis of dialkyl methylphosphonates, the order of reactivity was the following [92]: ^i^Pr > Me > Et ~ neopentyl

In contrast, applying base-catalyzed hydrolysis, the order of reactivity was the following [92]:Me > Et > ^i^Pr > neopentyl

In the case of diaryl esters, the rate constants were significantly higher. The rate is dependent on the electronic effects: electron-donating substituents slow down the hydrolysis [92]. This was also observed for cyclic phosphonates [123]. Ring cleavage also takes place during their alkaline hydrolysis [109].

The alkaline hydrolysis of phenyl methylphosphonate and *p*-nitrophenyl methylphosphonate (**74**) with the application of NaOH was also studied (Scheme 43) [124]. In this case, the nucleophilic attack of the hydroxide ion occurs on the phosphorus atom of the P=O function. The rate of the reaction increased with the increase of the hydroxide ion concentration.

Dimethyl 4-toluenesulfonyloxymethylphosphonate (**75**) was treated with 60% pyridine-H_2_O at room temperature to give the monomethyl derivative at an 82% yield (Scheme 44) [125]. The product was a good phosphorylating reagent to protect nucleosides.

The kinetics of the selective monohydrolysis of ethyl *p*-nitrophenyl chloromethylphosphonate (**77**) were studied in micellar solutions of a cationic surfactant (Scheme 45) [126,127].

In summary, the temperature, the solvent and the pH applied have a significant impact on the course of the alkaline hydrolysis. Considering the effect of the C-substituents attached to the P atom, the reactivity of the phosphonates decreases with increasing steric congestion, and increases due to the effect of electron-withdrawing substituents (e.g., in diethyl chloromethylphosphonate compared to diethyl methylphosphonate). Regarding the effect of the OR function, electron-withdrawing R groups (e.g., NO_2_Ph) increase the rate of the reaction. It is also noteworthy that the hydrolysis of aryl esters is faster than that of their aliphatic counterparts.

## 4. Dealkylation

### 4.1. Dealkylation of Phosphinates

In the case of certain ester groups, a high temperature treatment may also be effective to cleave the O-C bond. This is exemplified by the pyrolysis of diphenylphosphinates (**30**) with branched alkyl groups, affording the corresponding acid quantitatively (Scheme 46) [46]. This transformation required a treatment at 120–335 °C for 15 min, leaving diphenylphosphinic acid (**9**) in a solid form. The olefin that was liberated departed as a gas.

Dealkylations are most often performed using chemical agents, such as trimethylsilyl halides (TMSX). Following the fission, there is need for a treatment with water or methanol to form the corresponding acid [28]. A methyl-heptylphosphinic acid analogue of valproic acid (**80**) was prepared, in the hope of its bioactivity. In the final step of the synthesis, the resulting ethyl ester was cleaved by trimethylsilyl bromide (TMSBr) to give the target acid (**80**) (Scheme 47) [128]. The exact conditions were not reported.

Another example is the conversion of ethyl divinylphosphinate (**81**) to the corresponding acid (**83**) by treatment with TMSBr followed by methanolysis (Scheme 48) [129].

In addition to trimethylsilyl bromide, the iodide derivative (TMSI) is also a suitable dealkylating agent. Its application was demonstrated via the deethylation of a cyclic ethyl phosphinate (**83**) (Scheme 49) [130].

### 4.2. Dealkylation of Phosphonates

Certain phosphonates may also decompose on heating to give the corresponding acids. The di-*tert*-butyl esters (e.g., **85**), which may undergo dealkylation at 80 °C, are particularly suitable. The exact conditions were not reported in the example shown in Scheme 50 [79].

The dealkylation of the esters (**65**) may also take place with the aid of various reagents, most commonly with TMSX [131,132,133,134,135,136]. In the first step, the corresponding bis-trimethylsilyl ester (**87**) is formed, which provides the corresponding phosphonic acid (**66**) on treatment with water or methanol. This protocol, illustrated in Scheme 51, may be considered to be a gentle and convenient method [79,137]. 

Later on, the mechanism of the reaction was investigated, and it was found that TMSBr attacks the oxygen atom of the P=O function [138]. The sequence of the double dealkylation with TMSI is shown in Scheme 52 [139,140].

The cleavage with trimethylsilyl chloride (TMSCl) is not an often-used procedure [59]. This derivative is less reactive than TMSBr, and therefore requires a longer reaction time at a higher temperature. However, its lower cost and easier handling justifies its application. In a typical example, the mixture was heated up to 130–140 °C using three to four equivalents of TMSCl in chlorobenzene. The complete removal of the ester function took 8–36 h. The corresponding phosphonic acids (**66**) were obtained after hydrolysis (Scheme 53) [140]. 

The TMSC1 reagent can also be used in the preparation of PMEA/PMPA (**90**), which are known as antiviral agents (Scheme 54) [141].

McKenna compared the reactivity of TMSCl and TMSBr, and found that while the dealkylation with TMSCl, in most cases, was not complete within 1–9 days, the reaction with the bromide derivative was complete within 1–3 h (Table 6) [57].

The dealkylation reactions could be promoted by the addition of sodium iodide as a co-reagent to TMSCl [142]. In a few cases, the corresponding phosphonic acids (**66**) were obtained in good yields after treatment with this mixture of reagents at room temperature for 15–60 min (Scheme 55) [143].

Using lithium iodide, dialkyl phosphonates were cleaved under milder conditions [144]. In most of the cases, TMSBr was used in the dealkylations [145,146,147,148,149,150,151,152,153,154,155,156,157,158,159,160,161,162,163,164,165,166,167,168,169,170,171]. 

The conversion of phosphonates to the corresponding acids plays an important role in the synthesis of drugs. This is illustrated by the last step of the synthesis of Tenofovir (**90**) (Scheme 56) [172]. TMSCl/NaBr was applied in the preparation of other anti-HIV agents as well [173].

The double cleavage of the P(O)(O^i^Pr)_2_ function in a series of diisopropyl esters (**88**) was performed with TMSBr at 60 °C for 4–24 h. Using three equivalents of TMSBr, yields of around 80% were reported. A work-up including a treatment with NaOH and methanol gave the mono-Na salt of the phosphonic acid (**91**) (Scheme 57) [174].

Certain analogues of purine-based phosphonic acids are able to inhibit the FBSase enzyme, making them effective in the treatment of type 2 diabetes. During the synthesis of these type of compounds, the cleavage of the ester group was performed with TMSBr (Scheme 58) [175].

TMSBr may also be used in the dealkylation of metal complexes (**95**) (Scheme 59) [176].

In addition to TMSBr, boron tribromide was also proven to be an effective dealkylating reagent [177]. In this case, the reaction resulted in the formation of borophosphonate oligomers [–O–PR(O–)–O–B(O–)(O–)]_n_, along with the alkyl bromide by-product [53]. The methanolysis of the intermediate led to free phosphonic acid (**66**) and B(OMe)_3_ (Scheme 60).

In special cases, a cation exchange resin was used as a catalyst in dealkylations (Scheme 61) [178]. The phenylphosphonates (**88**) were reacted at 40 °C for a prolonged reaction time to afford phenylphosphonic acid (**66**) in variable yields, depending on the nature of the R group.

The dealkylation of diethyl ethylphosphonate (**65**) was performed using γ-alumina and silica gel. At a temperature of 300 °C, the cleavage of the P-C bond also occurred in addition to the desired fission of the C-O bond (Scheme 62) [179]. 

The monodealkylation of phosphonates (**98**) was also elaborated using sodium iodide in polar solvents (Scheme 63) [150,180,181,182].

The monodealkylation of phosphonates (**100**) was also performed under phase transfer catalytic conditions (Scheme 64) [183]. In this case, triethylbenzylammonium bromide or triethylbenzylammonium chloride (TEBAB or TEBAC, respectively) was applied as the catalyst in the dealkylation performed at reflux for 24–150 h.

Diethyl phosphonates could also be monodealkylated at 80–100 °C for good yields using lithium bromide or chloride [184]. There is an example of the use of lithium triethylborohydride as the monodealkylating agent in the preparation of GABA analogue phosphonic acids (**102**) (Scheme 65) [185].

In a few cases, the dealkylation could also be accomplished with amines, e.g., the monodealkylation of *H*-phosphonates was performed with *tert*-butylamine [54,55], but hydrazine could also be applied as a dealkylating agent [186].

## 5. Conclusions

*P*-acids have a significant role among drugs. Additionally, they have applications as herbicides and flame retardants as well. Hydrolysis is the most frequently used method to prepare *P*-acids that may be catalyzed by acids and bases. Comparing the two methods, it can be said that the alkaline hydrolysis takes place faster, and is less corrosive, but it is realized in two steps. The Na salt is formed in the first step, followed by the liberation of the free acid. While the acid-catalyzed hydrolysis may involve the cleavage of the P-O bond (A_Ac_2 mechanism) and that of the C-O bond (A_Al_1 mechanism) as well, the alkaline hydrolysis takes place via the cleavage of P-O bond. This explains why the isopropyl ester reacts faster than the methyl ester under acidic conditions, while in the base-catalyzed process, the former reacts considerably slower. The course of the reactions may be influenced by various factors, such as the temperature, solvent, pH, and the substituents attached to the phosphorus atom. The steric hindrance decreases the reaction rate, while the electron-withdrawing effects either in the C- or OC-substituents attached to the P atom increase the reactivity. Another method to convert the esters to *P*-acids is cleavage by trimethylsilyl halides. This approach is selective and requires milder conditions. In addition to hydrolysis under conventional conditions, MW-assisted variations were also elaborated, giving environmentally friendly and efficient accomplishments.

Through the examples discussed, it was shown that the hydrolysis of *P*-esters has not yet been explored fully, and generally that the reaction conditions have not been optimized. Despite these facts, especially on the basis of our own findings, this review may help us to get closer to a better understanding of this area of phosphorus chemistry. We advise the performance of the hydrolyses under acidic conditions using 0.5 mL of *cc*. HCl in 1 mL water to each 2 mmol of the phosphinate, or to each 1 mmol of the phosphonate.

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
