# Peer review of "The Hydrolysis of Phosphinates and Phosphonates: A Review"

_molecules, 2021, doi:10.3390/molecules26102840_

Round 1

Reviewer 1 Report

Review report “The Hydrolysis of Phosphinates and Phosphonates – 2 a review” by Nikoletta Harsági and György Keglevich, submitted to Molecules

      This paper is a nice review of our current knowledge on the patways to produce phosphinic and phosphonic acids and other biologically active compounds by hydrolysis or dealkylation. In general the manuscript is providing a detailed introduction and compilation of the reactions producing these compounds.

I missed in the introduction a few paragraphs describing the relevance of aqueous processes in the origin of organic complexity, just in an astrobiological context. For example, I think the authors could cite a few references concerning the relevance of phosphorus of the origin of life. On of them is Maciá (2005) that describes the relevance of phosphorus in chemical evolution, and that you might cite in several parts of your manuscript.

     The role of aqueous reactions and their role in the origin of life could be also emphasized. Some of these reactions are pre-terrestrial and already occurred in the parent bodies of undifferentiated meteorites, particularly carbonaceous chondrites (Rubin et al., 2007; Zolensky et al., 2008; Trigo-Rodríguez et al., 2019). The outcome of such aqueous alteration reactions in which P was probably associated with Fe-Ni metal alloys could have key relevance for the enrichment in organic complexity, particularly amino-acids in meteorites (Rotelli et al., 2016).

     At the end of the introduction please include a paragraph to organize and introduce the different sections to the reader. Please try to build a logical distribution of the sections, and try to focus the rest of the paper.

     The different chemical reactions are properly presented in several schemes. In any case, the manuscript lacks of a clear structure, and some examples of the implications in other areas. By doing that I think that the manuscript should be better organized, and increase its scope. It could be point out that the formation of phosphorus, fluorine and Na products by  alkaline hydrolysis could be also relevant in the context of comets (Gardner et al., 2020 and references therein).

        Finally, I found that the conclusions’ section should be better organized. Please enumerate your findings and answer: what are the main implications of them? Please also try to emphasize what is really novel compared with previous work.

       In consequence, I think that this manuscript should be revised before being considered for publication. My recommendation is moderate review, and I’d like to see the revised version of the ms.

            References

            Gardner E. Et al. (2020) The detection of solid phosphorus and fluorine in the dust from the coma of comet 67P/Churyumov-Gerasimenko. Monthly Notices of the Royal Astronomical Society, Volume 499, 1870-1873

Maciá E. (2005) The role of phosphorus in chemical evolution. Chem. Soc. Rev., 2005, 34(Advance Article), DOI: 10.1039/b416855k

            Rotelli L. et al. (2016) The key role of meteorites in the formation of relevant prebiotic molecules in a formamide/water environment, Nature Sci. Rep. 6:38888, DOI: 10.1038/srep38888

            Rubin A.E., et al. (2007) Progressive aqueous alteration of CM carbonaceous chondrites. Geochimica et Cosmochimica Acta 71, 2361-2382.

Trigo-Rodríguez, J.M. et al. (2019) Accretion of water in carbonaceous chondrites: current evidence and implications for the delivery of water to early Earth, Space Science Reviews 215:18, 27 pp.

Zolensky, M.E., Krot, A.N., Benedix, G. (2008): Record of low-temperature alteration in asteroids.  Pp. 429–462 in: Oxygen in the Solar System (G.J. MacPherson, D.W. Mittlefehldt, J.H. Jones & S.B. Simon, editors). Reviews in Mineralogy & Geochemistry, 68. Mineralogical Society of America, Washington, D.C.

Author Response

Dear senior Referee 1:

Thanks for scrutinizing our ms and for your kind words and bettering ideas. 

This paper is a nice review of our current knowledge on the patways to produce phosphinic and phosphonic acids and other biologically active compounds by hydrolysis or dealkylation. In general the manuscript is providing a detailed introduction and compilation of the reactions producing these compounds.

I missed in the introduction a few paragraphs describing the relevance of aqueous processes in the origin of organic complexity, just in an astrobiological context. For example, I think the authors could cite a few references concerning the relevance of phosphorus of the origin of life. On of them is Maciá (2005) that describes the relevance of phosphorus in chemical evolution, and that you might cite in several parts of your manuscript.

Well, the relevance of P in the origin of life is indeed interesting, but its discussion would not be relevant in our review focused on the hydrolysis of P-esters. I ask you to disregard citing Macia.

     The role of aqueous reactions and their role in the origin of life could be also emphasized. Some of these reactions are pre-terrestrial and already occurred in the parent bodies of undifferentiated meteorites, particularly carbonaceous chondrites (Rubin et al., 2007; Zolensky et al., 2008; Trigo-Rodríguez et al., 2019). The outcome of such aqueous alteration reactions in which P was probably associated with Fe-Ni metal alloys could have key relevance for the enrichment in organic complexity, particularly amino-acids in meteorites (Rotelli et al., 2016).

O, yes you are right, the role of aqueous reactions should have been pointed out. In the revised version we did so and inserted a paragraph below Figure 1.

"For a long time, water was used as solvent in only hydrolyses. Despite its favorable properties (cheap, available, safe and “green”), water could not spread as a general solvent. This is due to the low solubility of organic substrates. The application of co-solvents, such as alcohols, DMF, acetone and acetonitrile is a good possibility. However, the regeneration of water or water-solvent mixtures is not easy."

     At the end of the introduction please include a paragraph to organize and introduce the different sections to the reader. Please try to build a logical distribution of the sections, and try to focus the rest of the paper.

This request was obeyed. We inserted the following segment at the end of the Introduction to guide the readers:

"In this survey, we discuss the acidic hydrolysis of phosphinates and phosphonates. This is followed by the presentation of the alkaline and basic hydrolysis of phosphinates and that of phosphonates. The reactivity of the different substrates, the effect of the substituents, and green chemical aspects are in the focus. Last but not least, the conversion of P-esters to acids by dealkylation is summarized."

     The different chemical reactions are properly presented in several schemes. In any case, the manuscript lacks of a clear structure, and some examples of the implications in other areas. By doing that I think that the manuscript should be better organized, and increase its scope. It could be point out that the formation of phosphorus, fluorine and Na products by  alkaline hydrolysis could be also relevant in the context of comets (Gardner et al., 2020 and references therein).

We beleive that the formation of P and F along with Na products is not too relevant in our paper. Pls allow not to include this!

        Finally, I found that the conclusions’ section should be better organized. Please enumerate your findings and answer: what are the main implications of them? Please also try to emphasize what is really novel compared with previous work.

The Conclusions part was well completed:

"Despite these facts, especially on the basis of our own findings, this review may help to get closer to a better understanding of this area of phosphorus chemistry. It can be advised to perform the hydrolyses under acidic conditions using 0.5 mL of cc. HCl in 1 mL of water to each 2 mmol of the phosphinate, or to each 1 mmol of the phosphonate."

       In consequence, I think that this manuscript should be revised before being considered for publication. My recommendation is moderate review, and I’d like to see the revised version of the ms.

Thanks for the minor revision suggestion and for providing us with interesting references. They surely will be read to our training.

The new parts were highlighted in yellow in the revised ms.          

Reviewer 2 Report

This is an excellent and comprehensive review of an important topic, written clearly and authoritatively by a group who have been active in the area. It is a valuable contribution to the literature of organophosphorus chemistry and can be published essentially without change, subject only to correction of a handful of very minor errors. The authors are to be congratulated.

Required changes:

Page 2, line 61 - change 'despite of this, the application of hydrochloric acid was spread' to 'despite this, the application of hydrochloric acid was widespread'

Page 3/4, line 102 and 114 - change to 'methyl dialkylphosphinates' (no space in second word)

Page 23, Scheme 42 - enzyme name needs changed into English

Ref 72 change 'hidrolysing' to 'hydrolysing'

Ref 103 - change 'phosphsrus' to 'phosphorus'

Author Response

Dear senior Referee 2:

Let us thank you for your favorable opinion and advice.

You required the following changes:

Page 2, line 61 - change 'despite of this, the application of hydrochloric acid was spread' to 'despite this, the application of hydrochloric acid was widespread'

Page 3/4, line 102 and 114 - change to 'methyl dialkylphosphinates' (no space in second word)

Page 23, Scheme 42 - enzyme name needs changed into English

Ref 72 change 'hidrolysing' to 'hydrolysing'

Ref 103 - change 'phosphsrus' to 'phosphorus'

All of your suggestions were obeyed. Thansk for your care.

The corrected parts were highlighted in yellow.

Round 2

Reviewer 1 Report

Thanks for the clarifications added